# I Choose You: Selecting Accurate Reference Genes for qPCR Expression Analysis in Reproductive Tissues in *Arabidopsis thaliana*

**DOI:** 10.3390/biom13030463

**Published:** 2023-03-02

**Authors:** Maria João Ferreira, Jessy Silva, Sara Cristina Pinto, Sílvia Coimbra

**Affiliations:** 1LAQV/REQUIMTE, Biology Department, Faculty of Sciences, University of Porto, Rua do Campo Alegre s/n, 4169-007 Porto, Portugal; 2School of Sciences, University of Minho, Campus de Gualtar, 4710-057 Braga, Portugal

**Keywords:** reference genes, reproductive tissues, expression analysis, normalisation, qPCR

## Abstract

Quantitative real-time polymerase chain reaction (qPCR) is a widely used method to analyse the gene expression pattern in the reproductive tissues along with detecting gene levels in mutant backgrounds. This technique requires stable reference genes to normalise the expression level of target genes. Nonetheless, a considerable number of publications continue to present qPCR results normalised to a single reference gene and, to our knowledge, no comparative evaluation of multiple reference genes has been carried out in specific reproductive tissues of *Arabidopsis thaliana*. Herein, we assessed the expression stability levels of ten candidate reference genes (*UBC9*, *ACT7*, *GAPC-2*, *RCE1*, *PP2AA3*, *TUA2*, *SAC52*, *YLS8*, *SAMDC* and *HIS3.3*) in two conditional sets: one across flower development and the other using inflorescences from different genotypes. The stability analysis was performed using the RefFinder tool, which combines four statistical algorithms (geNorm, NormFinder, BestKeeper and the comparative ΔCt method). Our results showed that *RCE1*, *SAC52* and *TUA2* had the most stable expression in different flower developmental stages while *YLS8*, *HIS3.3* and *ACT7* were the top-ranking reference genes for normalisation in mutant studies. Furthermore, we validated our results by analysing the expression pattern of genes involved in reproduction and examining the expression of these genes in published mutant backgrounds. Overall, we provided a pool of appropriate reference genes for expression studies in reproductive tissues of *A. thaliana*, which will facilitate further gene expression studies in this context. More importantly, we presented a framework that will promote a consistent and accurate analysis of gene expression in any scientific field. Simultaneously, we highlighted the relevance of clearly defining and describing the experimental conditions associated with qPCR to improve scientific reproducibility.

## 1. Introduction

*Arabidopsis thaliana*, like all land plants, developed a life cycle that alternates between a haploid gametophytic generation and a diploid sporophytic generation [1]. The sporophyte generates two types of spores: microspores and megaspores, giving rise to the male and female gametophytes, respectively [2,3]. The male gametophyte, or pollen grain, comprises a vegetative cell that will develop into a pollen tube (PT), and a generative cell, which through a mitotic division, will form the two male gametes [4]. The female gametophyte, also known as embryo sac, exhibits a Polygonum-type pattern [5] composed of two synergid cells, a central cell, egg cell, and three antipodals [6,7]. When the pollen grain and embryo sac are mature, a unique process to Angiosperms occurs called double fertilisation. This event results in seed development and consists of the fusion of a male gamete with the egg cell, originating the diploid zygote, and the fusion of the other male gamete with the central cell, generating the triploid endosperm [8].

Several players have been reported to participate in the successful establishment of a new sporophytic generation, with many showing very specific spatiotemporal gene expression patterns [9]. This is the case of arabinogalactan proteins (AGPs), a protein family known for the high level of glycosylation in its members [10]. For example, AGP6 and AGP11, specifically expressed in the stamen and pollen, are thought to be involved in pollen grain release and PT endosome machinery, since knockout and knockdown mutations of these genes cause pollen abortion [11,12,13]. On the other hand, *JAGGER*, an AGP highly expressed in female tissues, was reported to be involved in the signalling pathway that blocks PT attraction once fertilisation occurs in *jagger* mutants, the persistent synergid does not degenerate after double fertilisation and continues to attract more PTs into the embryo sac (polytubey phenotype; [14]).

The study of plant reproduction is very important as it will provide knowledge to develop tools that will aid in maintaining successful fertilisation under stressful environmental conditions as well as in the production of resilient seeds. It is, therefore, necessary to discover all components underlying the pathways of seed formation and development. Monitoring gene expression levels on the different reproductive tissues (sporophytic and gametophytic), as well as in distinct genotypes [mutant (mt) versus wild type (wt)] is a common practice among plant reproduction studies. Spatiotemporal gene expression is preferably measured by quantitative polymerase chain reaction (qPCR) due to its sensitivity, accuracy, and cost-effectiveness [15]. Nevertheless, RNA purity, cDNA synthesis, primers efficiency, selection of internal controls, and statistical methods for analysis are variables inherent to qPCR technique that must be addressed [16,17]. The relative expression ratio [18], which compares the target gene transcripts to internal controls, is still considered the most adequate method for sample normalisation, minimizing the artefactual variability of total cDNA abundance among samples. Therefore, the choice of reference genes for experiments is fundamental in correctly interpreting the results [19]. A reference gene should be stably expressed on various cells/tissues under analysis as well as in different experimental conditions/treatments [16,19,20]. In the pre-genomic era, reference genes included families considered to be essential for a correct cell function, such as actin (ACT), ubiquitin (UBQ) and alpha-tubulin (TUA) [21,22,23]. However, this assumption was shown to be wrong for both animal and plant organisms [24,25,26,27]. For example, [28] time-course transcriptome analysis of Arabidopsis siliques showed *ACT2*, *TUB2* and *TUB6* as down-regulated genes in a mutant background, revealing that these genes were inappropriate internal controls for this specific qPCR experimental condition. To date, no validation of reference genes on *A. thaliana* reproductive tissues across different developmental stages has been described.

Here we report the expression stability of ten candidate reference genes [*UBIQUITIN CONJUGATING ENZYME 9* (*UBC9*), *ACTIN 7* (*ACT7*), *GLYCERALDEHYDE-3-PHOSPHATE DEHYDROGENASE C2* (*GAPC-2*), *RUB1 CONJUGATING ENZYME 1* (*RCE1*), *PROTEIN PHOSPHATASE 2A SUBUNIT A3* (*PP2AA3*), *TUBULIN ALPHA-2 CHAIN* (*TUA2*), *SUPPRESSOR OF ACAULIS 52* (*SAC52*), *YELLOW-LEAF-SPECIFIC GENE 8* (*YLS8*), *S-ADENOSYLMETHIONINE DECARBOXYLASE* (*SAMDC*) and *HISTONE 3.3* (*HIS3.3*)] in different reproductive tissues at different stages of development and with inflorescences of wt and mt of A. thaliana, using RefFinder [29] which integrates four statistical programs, namely geNorm [19], NormFinder [30], BestKeeper [31], and the comparative ΔCt method [32]. Previously reported genes involved in reproduction were used to validate the stability of the reference genes. Our study provides guidance to other scientists on how to choose reference genes for further studies using *A. thaliana* reproductive tissues.

## 2. Materials and Methods

### 2.1. Plant Material and Growth Conditions

Wild-type (wt) *A. thaliana* (L.), Heynh. Columbia-0 (Col-0), and Nossen-0 (No-0) seeds were obtained from Nottingham Arabidopsis Stock Centre (NASC). The *jagger1-2* (GABI-Kat 134A10) in Col-0 background and *agp6 agp11* (Ds54-4754-1 and Ds11-4025-1) in No-0 background mutant lines were previously characterised by [14] and [12], respectively. *A. thaliana* seeds were sown directly in soil (COMPO SANA^®^, COMPO, Münster, Germany) and kept for 48 h at 4 °C in the dark to induce stratification. Afterwards, all plants were grown under long-day conditions (16 h light at 22 °C and 8 h darkness at 18 °C) with 50–60% relative humidity and light intensity at 180 µmol m^−2^ s^−1^.

### 2.2. Selection of Candidate Reference Genes and Primer Design

A list of commonly used reference genes was obtained in a literature search. We analysed several published studies about the selection of reference genes for gene expression experiments using several tissues, conditions and plant species, besides *A. thaliana*. We then evaluated their expression on the *A. thaliana* reproductive tissues using the transcriptomic database EVOREPRO (https://evorepro.sbs.ntu.edu.sg/, accessed on 15 November 2021) [33], and selected ten genes from different functional classes that are essential to the correct function of a cell and which expression was high and stable across different tissues: male (microspore, bicellular pollen, tricellular pollen, mature pollen, pollen tube, generative cell, and sperm cell); female (ovary, ovule, and egg cell); flower (carpels, stigmatic tissue, stamen filaments, anthers, petals, sepals, flower buds, and receptacles); and seeds (endosperm, young seed, seed, and germinating seed) defined by [33] (Appendix A).

The coding sequences of the candidate genes were retrieved from the National Center for Biotechnology Information (NCBI, http://www.ncbi.nlm.nih.gov/, accessed on 15 November 2021). The candidate reference genes were: *UBC9* (AT4G27960), *ACT7* (AT5G09810), *GAPC-2* (AT1G13440), *RCE1* (AT4G36800), *PP2AA3* (AT1G13320), *TUA2* (AT1G50010), *SAC52* (AT1G14320), *YLS8* (AT5G08290), *SAMDC* (AT3G02470) and *HIS3.3* (AT4G40030). Primers were designed using Primer3 v.4.1.0 [34,35,36] according to the following parameters: annealing preferably on the 3′ of the transcript, primer length between 18–25 bp, GC content between 40–60%, melting temperature around 60 °C, and amplicon size between 70 and 150 bp. Structural aspects of the primers were also criterion of primer design: primers with G or C repeats longer than three bases, with more than two G or C in the last 5 bases at the 3′ end, and with long (>4) repeats of any nucleotide were avoided; primers with G and C at the ends were chosen when possible (Table 1).

The target genes *AGP6*, *AGP11* and *JAGGER* and the respective qPCR primers were previously published [14,37] (Appendix A). The specificity of the primers was confirmed by a conventional PCR and electrophoresis on 1% (*w*/*v*) agarose gel. The amplified products were purified using the GeneJET PCR Purification Kit (Thermo Fisher Scientific, Waltham, MA, USA) and sequenced by Eurofins Genomics (Ebersberg, Germany) to confirm the identity of the PCR product.

### 2.3. Sample Collection, RNA Isolation and cDNA Synthesis

A total of 33 samples for RNA extraction were collected from five-week-old plants, representing 11 tissues from three biological samples. Samples were independently prepared and divided into two experimental sets: flower development and genotype. The flower development group contained three biological replicates of seven different Col-0 reproductive structures, at different developmental stages according to [38]: flowers stage 1 to 6—at which flower primordium is formed; flowers stage 7 to 12—when both male and female gametophytes develop; flowers stage 13 to 14—during which double fertilisation occurs; flowers stage 15 to 16—when the zygote starts to develop within the seed; pistils from stage 12 flowers—as an example of a female tissue containing mature ovules; anthers from stage 12 flowers—as an example of a male tissue containing mature pollen grains; and siliques from stage 17—harbouring mature seeds, as an example of a post-fertilisation tissue [38,39]. Together, these corresponded to a total of 21 samples (Appendix A). The genotype set included three biological replicates of inflorescences—the most commonly used sample type in plant reproduction when confirming the expression of a mutated gene—from wt Col-0, wt No-0, *agp6 agp11* and *jagger*, totalling 12 samples. Samples were immediately frozen in liquid nitrogen and stored at −80 °C for follow-up experiments. All images from the different stages of flower development and tissues of *A. thaliana* used in this study were acquired by a SMZ168 Stereo Zoom microscope (Motic, Barcelona, Spain) and images were captured using a Moticam 2500 (Motic, Barcelona, Spain) camera and the Motic Images Plus 2.0 (Motic) software. Total RNA was extracted using the RNeasy Plant Mini Kit (QIAGEN, Manchester, UK) as per the manufacturer’s protocol. RNA purity and concentration were measured using a spectrophotometer (DS-11 Series Spectrophotometer/Fluorometer, DeNovix, Wilmington, DE, USA). Only RNA samples with absorption ratios of 1.8–2.1 at 260/280 nm and around 2.0 for 260/230 nm were used for further analysis. RNA integrity was evaluated by checking the presence of the 25S and 18S ribosomal RNAs bands in a 1% (*w*/*v*) agarose gel electrophoresis. Following the manufacturer’s instructions, 150 ng or 1 μg of total RNA from reproductive structures or inflorescences, respectively, was treated with DNase I, RNase-free (Thermo Fisher Scientific, Waltham, MA, USA). cDNA was synthesized using SuperScript™ III First-Strand Synthesis System (Invitrogen, Waltham, MA, USA) and oligo(dT)_20_ primers to initiate the reactions, according to the manufacturer’s guidelines. The cDNA products of reproductive tissues and inflorescences were diluted at 1.25 ng/μL and 4 ng/μL, respectively, with nuclease-free water before qPCR.

### 2.4. qPCR Analysis

qPCR reactions were performed in a 10 µL final volume containing 5 µL of 2× SsoAdvanced^TM^ Universal SYBR^®^ Green Supermix (Bio-Rad, Hercules, CA, USA), 0.125 µL of each specific primer pair at 250 nM, 0.75 µL of nuclease-free water and 4 µL of diluted cDNA template. The reactions were performed in 96-well plates and run in a CFX96 Real-Time System (Bio-Rad, USA) with the following cycling conditions: initial denaturation at 95 °C for 30 s, followed by 40 cycles of 95 °C for 15 s, and 60 °C for 30 s, and any additional data acquisition step of 15 s at the optimal acquisition temperature. All reactions were run in three technical replicates and all assays included non-template controls (NTCs). Using three points of a tenfold dilution series (1:10, 1:100 and 1:1000) from pooled cDNA (wt inflorescences and siliques), a standard curve was generated to estimate the PCR efficiency of each primer pair using CFX Maestro software v. 2.0 (Bio-Rad, USA). The slope and the coefficient of determination (R^2^) were obtained from an equation of the linear regression line and the amplification efficiency (E) was calculated according to the formula E = (10^−1/slope^ − 1) × 100%. The R^2^ value should be higher than 0.980 with the E value between 90% and 110%. After completion of the amplification reaction, melt curves were generated by increasing the temperature from 65 to 95 °C with fluorescence readings acquired at 0.5 °C increments. From the melt curve, the optimal temperature for data acquisition (3 °C below the melting temperature of the specific PCR product) was determined and the specificity of primers was confirmed. The sample maximization method was employed as the run layout strategy, in which all samples for each defined set were analysed in the same run and, thus, different genes were analysed in distinct runs [20]. The quantitative cycle (Cq), baseline correction, and threshold setting were automatically calculated by the CFX Maestro software v. 2.0 (Bio-Rad, Hercules, CA, USA). The products of qPCR were verified through 1% (*w*/*v*) agarose gels.

### 2.5. Expression Stability Analysis of Candidate Reference Genes

The mean Cq of ten potential reference genes in all 33 samples were introduced in the online software RefFinder (https://www.heartcure.com.au/reffinder/, accessed on 30 November 2021; [29]) to evaluate the gene expression stability. By integrating four different statistical algorithms, namely geNorm [19], NormFinder [30], BestKeeper [31] and the comparative ΔCt method [32], the software analyses the ranking of each algorithm for each gene and engenders an overall final ranking list of reference genes according to their stability. The geNorm algorithm calculates the expression stability value (M) of candidate reference genes, considering the average pairwise variation of each gene with all other control genes. A small M value indicates a more stable performance of the gene. The geNorm programme also calculates the pairwise variation (V_n_/V_n+1_, n represents the number of reference genes) to determine the optimal number of reference genes necessary for precise qPCR data normalisation. A V_n_/V_n+1_ value below 0.15 indicates that n is appropriate to normalise data and there is no need to include an additional reference gene [19]. NormFinder entails an ANOVA-based model to assess inter and intra-group variation, and the most stable gene possesses the lowest stability value [30]. As for BestKeeper, the programme investigates the candidate genes stability based on the standard deviation (SD) and coefficient of variation (CV) of the Cq values in a sample pool. The lower the SD ± CV, the more stable is the gene [31]. Finally, the comparative ΔCt method uses the SD to estimate the stability of all reference genes, with the most stable one having a lower SD [32]. Furthermore, RefFinder produces a comprehensive ranking of the expression stability of the candidate reference genes in the four different programs by calculating the geometric mean of their weights for the overall final ranking [29].

### 2.6. Validation of Reference Genes by Expression Analysis of AGP Genes

The reliability of the reference genes was validated using three target genes, namely *AGP6*, *AGP11* and *JAGGER*. The expression levels of the target genes were quantified in all samples using the three most stable, the two most stable and the two least stable reference genes. The qPCR reactions were conducted as described above. The relative gene expression was calculated using the 2^−ΔΔCt^ method [40]. Data were statistically treated using GraphPad Prism 8 software (www.graphpad.com, accessed on 15 December 2021). For each analysis, relative expression differences were compared using a two-way ANOVA followed by Dunnett’s multiple comparisons test. Statistical significance was considered at α = 0.05.

## 3. Results

### 3.1. Selection of Reference Genes

To identify the best reference genes for qPCR normalisation in gene expression studies regarding reproductive tissues, ten candidate reference genes (*UBC9*, *ACT7*, *GAPC-2*, *RCE1*, *PP2AA3*, *TUA2*, *SAC52*, *YLS8*, *SAMDC* and *HIS3.3*) were selected, based on previous reports of qPCR studies in plants [27,41,42,43,44,45,46,47]. The selected genes represent distinct biological pathways to avoid co-regulation. Involved in post-translational modifications, *UBC9* is necessary for SUMOylation [48], *PP2AA3* participates in phosphorylation [49] and *RCE1* performs neddylation [50]. Other reference genes are constituents of the cytoskeleton [51] and microtubules [52], such as *ACT7* and *TUA2*. Part of metabolic pathways, *GAPC-2* is implicated in glycolysis [53] and *SAMDC* is part of the polyamine biosynthesis pathway [54]. Finally, chosen reference genes comprise ribosomal and nuclear proteins involved in DNA replication (*YLS8*) [55], gene regulation (*HIS3.3*) [56], and translation regulation (*SAC52*) [57].

The biological processes mentioned above are considered fundamental for the normal function of a cell, thus they should be highly expressed in all cells. Nonetheless, to confirm that the ten candidate reference genes were highly expressed on the reproductive tissues used in this study, we analysed their expression levels in the transcriptomic database EVOREPRO [33] (Appendix A). Overall, the selected genes presented high expression values in the different tissues analysed.

### 3.2. PCR Amplification Efficiency and Specificity

The primers of each reference gene were tested in a conventional PCR and the amplicon lengths were checked in a 1% (*w*/*v*) agarose gel, which showed one single band with the expected fragment sizes (72–141 bp) (Table 1) and without primer dimers or any impurities. The amplified PCR products were sequenced to confirm the identity of the PCR amplicon. Standard curves for each primer pair were generated to confirm the specificity of primers. All products exhibited only a single amplification peak in the melt curve representative of a single product amplification (Appendix A). The R^2^ of the standard curves ranged from 0.989 for *SAMDC* to 1.000 for *UBC9* (Table 2), demonstrating the linearity of the Cq values with the logarithmic of the starting quantity. The amplification E varied from 90% for *YLS8* to 107.5% for *SAMDC* (Table 2).

### 3.3. Cq Value Distribution of the Candidate Reference Genes

The distribution of Cq values obtained in qPCR data of the ten candidate reference genes is shown in the box plots (Figure 1). The Cq values ranged from 19.22 to 27.66 in the flower development set (Figure 1A), and from 17.67 to 25.89 in the genotype set (Figure 1B). This demonstrates the narrow range of gene expression level across all samples. As the transcript levels are negatively correlated with Cq values, *SAMDC* revealed the highest expression, having the lowest mean Cq values in the flower development (19.22) and genotype (17.67) sets. *GAPC*-2, *TUA2* and *SAC52* displayed high expression in both biological sets. Regarding flower development set (Figure 1A), the median Cq values were 21.13, 21.64 and 21.68, respectively. In the genotype set, *GAPC*-2, *TUA2* and *SAC52* exhibited median Cq values of 18.74, 19.33 and 19.05, respectively (Figure 1B). On the other hand, *PP2AA3* had the lowest relative expression both for flower development (mean Cq value of 25.55) and genotype set (mean Cq value of 23.14).

### 3.4. Expression Stability Ranking of the Candidate Reference Genes

The analysis of the expression stability levels of the ten candidate reference genes was carried out using the RefFinder [29]. This tool integrates four different algorithms to calculate the stability levels of reference genes: geNorm [19], NormFinder [30], BestKeeper [31] and the comparative ΔCt method [32]. A top-ranking table of the most stable reference genes for each set of samples according to each statistical program is on Appendix A. The analysis was performed in the two experimental sets, flower development and genotype, which included all 21 and 12 samples, respectively. Additionally, specific subsets of the flower development set were also analysed to study in more detailed tissues and developmental stages such as flower (st 1–16), that included flower st 1–6, flower st 7–12, flower st 13–14 and flower st 15–16, silique (st 17), pistil (st 12) and anther (st 12). Similarly, two subsets of the genotype set that represent different *A. thaliana* ecotypes were also analysed, namely wt Col-0 vs. *jagger* and wt No-0 vs. *agp6 agp11*.

In the geNorm analysis, the expression stability of the ten candidate reference genes was assessed by calculating the M value in the RefFinder tool (Figure 2). The M value is the mean variation of a gene relative to all other genes. The lower the M value, the more stable is the gene [19]. The ten reference genes across all experimental sets had M values below 0.8, suggesting that these genes were stable across all samples (Figure 2).

For the flower development set, *TUA2* and *RCE1* were the two most stable genes with a M value of 0.44, whilst *SAMDC* (0.74) was the least stable gene (Figure 2A). *YLS8* and *PP2AA3* (0.39) ranked the highest stability in flower (st 1–16) samples (Figure 2B), while *TUA2*, *YLS8*, *PP2AA3*, *SAC52*, *HIS3.3* and *ACT7* (0) were the most stable combination in silique (st 17) samples (Figure 2C). *TUA2*, *RCE1*, *YLS8* and *HIS3.3* (0) were the most stable genes in pistil samples (Figure 2D) and, when anther samples were considered, *TUA2*, *RCE1*, *UBC9*, *YLS8*, *SAC52*, *SAMDC* and *ACT7* (0) were ranked as the most stable genes (Figure 2E). In contrast, *UBC9* was the least stable gene in flower (st 1–16) (Figure 2B) and pistil (st 12) (Figure 2D) subsets, with a M value of 0.61 and 0.43, respectively. *GAPC-2* was the least stable gene in silique (st 17) (Figure 2C) and anther (st 12) (Figure 2E) subsets, with a M value of 0.39 and 0.27, respectively (Figure 2A). *YLS8* was one of the most stable genes across all subsets (Figure 2A).

In the genotype set, *HIS3.3* and *ACT7* were the two most stable genes with the lowest M value of 0.39, while *GAPC-2* (0.56) was the least stably expressed gene (Figure 2F). In addition, *HIS3.3* and *SAMDC* (0) were the most suitable genes in wt Col-0 vs. *jagger* samples (Figure 2G), whilst *HIS3.3*, *YLS8* and *GAPC-2* (0) were identified as the most stable reference genes in wt No-0 vs. *agp6 agp11* samples (Figure 2H). *GAPC-2* (0.57) and *SAMDC* (0.47) were the least stable genes in wt Col-0 vs. *jagger* (Figure 2G) and wt No-0 vs. *agp6 agp11* (Figure 2H) subsets, respectively.

geNorm also calculates the optimal number of reference genes essential for accurate qPCR normalisation by calculating the pairwise variation (V_n_/V_n+1_, n represents the number of reference genes). For a V_n_/V_n+1_ value below 0.15, no additional reference gene is required to be included in the analysis [19]. All combinations of genes presented a V_2_/V_3_ below 0.15 in both sets and all subsets, indicating that two reference genes are adequate for normalisation of these datasets (Figure 3).

### 3.5. NormFinder

The expression stability values were calculated by NormFinder in the RefFinder software (Figure 4). The expression stability value is based on inter and intra-group variations and the lowest value indicates a more stable gene [30].

Among the flower development set, *RCE1* and *SAC52* were the top ranked genes with the lowest stability values of 0.32 and 0.36, respectively, while *SAMDC* (0.80) was the least stable gene (Figure 4A). *YLS8* (0.27) and *RCE1* (0.31) ranked the highest in flower (st 1–16) samples (Figure 4B), while *SAC52*, *YLS8*, *TUA2*, *ACT7*, *HIS3.3* and *PP2AA3* (0.12) were identified as the best reference genes in silique (st 17) samples (Figure 4C). Meanwhile, *RCE1*, *SAC52*, *YLS8*, *TUA2* and *SAMDC* ranked highest in both pistil (st 12) (Figure 4D) and anther (st 12) (Figure 4E) subsets, with a stability value of 0.30 and 0.17, respectively (Figure 4A). Similarly, *HIS3.3*, *PP2AA3*, and *GAPC-2* (0.30) were also the most stable genes in pistil (st 12) samples (Figure 4D), while *ACT7* and *UBC9* (0.17) were top ranked genes in anther (st 12) samples (Figure 4E). In contrast, *UBC9* (0.63) was the most unstable gene in flower (st 1–16) (Figure 4B), while *GAPC-2* (0.62) was the least stable gene in silique (st 17) samples (Figure 4C). *UBC9* and *ACT7* (0.54) were the least stable genes in the pistil (st 12) subset (Figure 4D), while *GAPC-2*, *PP2AA3* and *HIS3.3* (0.44) were the least stable genes in the anther (st 12) subset (Figure 4E).

Additionally, *YLS8* and *HIS3.3* were the two most stable reference genes in the genotype set, with the lowest stability values of 0.25 and 0.29, respectively, whilst *GAPC-2* was the most unstable gene with the highest expression stability value of 0.50 (Figure 4F). *HIS3.3* and *SAMDC* (0.25) were the most stable genes in wt Col-0 vs. *jagger* samples (Figure 4G), while *HIS3.3*, *YLS8* and *GAPC-2* (0.23) were the three most stable genes in wt No-0 vs. *agp6 agp11* samples (Figure 4H). *GAPC-2* (0.64) was the least stable gene in wt Col-0 vs. *jagger* subset (Figure 4G), and as for wt No-0 vs. *agp6 agp11* subset, *SAMDC* (0.50) was the least stable gene (Figure 4H).

### 3.6. BestKeeper

BestKeeper programme was used to calculate the expression stability value by computing the SD of the mean Cq values (Figure 5). A lower SD reflects a more stable reference gene [31].

In the flower development set, BestKeeper inferred that *UBC9* and *GAPC-2* were the most stable reference genes, with a SD value of 0.52 and 0.57, respectively, while *SAMDC* (1.02) was the least stable gene (Figure 5A). *GAPC-2* (0.33) and *UBC9* (0.56) were the most stable genes in flower (st 1–16) samples (Figure 5B), whilst *UBC9* and *RCE1* (0) were the most stable genes in silique (st 17) samples (Figure 5C). *UBC9* (0.44) was the most stable gene in pistil (st 12) samples (Figure 5D) and, together with *RCE1*, *TUA2*, *SAC52*, *YLS8*, *SAMDC* and *ACT7* (0) were found to be the most stable genes in anther (st 12) samples (Figure 5E). *TUA2* (0.96) was the worst gene in flower (st 1–16) (Figure 5B), while *GAPC-2* (0.67) was the least stable gene in silique (st 17) samples (Figure 5C). *ACT7* (1.11) was the least stable gene in the pistil (st 12) subset (Figure 5D), while *GAPC-2*, *PP2AA3* and *HIS3.3* (0.44) were the least stable genes in the anther (st 12) subset (Figure 5E).

*UBC9* and *RCE1* were the most stable genes in the genotype set, with the lowest SD values of 0.65 and 0.75, respectively, while *SAC52* (1.11) was the least stable gene (Figure 5F). The most stable genes in wt Col-0 vs. *jagger* samples were *UBC9* (0.56) and *GAPC-2* (0.67) (Figure 5G), and for wt No-0 vs. *agp6 agp11* samples the programme identified *YLS8*, *GAPC-2* and *HIS3.3* (0.28) as the most stable ones (Figure 5H). *SAC52* (1.22) was the least stable gene in wt Col-0 vs. *jagger* subset (Figure 5G), while *SAC52* and *SAMDC* (0.67) were the least stable genes in wt No-0 vs. *agp6 agp11* subset (Figure 5H).

### 3.7. *Δ*Ct Method

The comparative ΔCt method calculates the average of SD to determine the expression stability of the reference genes (Figure 6). The most stable gene has the lower SD value [32].

Among the flower development group, *RCE1* and *SAC52* were the top two ranked genes, with an average SD of 0.63 and 0.64, while *SAMDC* (0.92) was the least stable gene (Figure 6A). *YLS8* (0.52) and *RCE1* (0.54) were the most stable genes in flower (st 1–16) samples (Figure 6B), while *YLS8*, *SAC52*, *TUA2*, *HIS3.3*, *PP2AA3* and *ACT7* (0.26) were the most stable genes in silique (st 17) samples (Figure 6C). *YLS8*, *SAC52*, *TUA2*, *HIS3.3*, *PP2AA3*, *GAPC-2*, *RCE1 and SAMDC* (0.38) were the most stable genes in pistil (st 12) samples (Figure 6D), whilst *YLS8*, *SAC52*, *TUA2*, *RCE1*, *SAMDC*, *ACT7* and *UBC9* (0.19) were the most stable genes in anther (st 12) samples (Figure 6E). *UBC9* (0.75) was the worst gene in flower (st 1–16), while *GAPC-2* (0.67) was the least stable gene in silique (st 17) samples (Figure 6C). *UBC9* and *ACT7* (0.62) were the least stable genes in the pistil (st 12) subset (Figure 6D), while *GAPC-2*, *PP2AA3* and *HIS3.3* (0.45) were the least stable genes in the anther (st 12) subset (Figure 6E). 

*YLS8* and *HIS3.3* were the most stable genes in the genotype set, with the expression stability value of 0.48 and 0.50, respectively, while *GAPC-2* (0.62) was the most unstable one (Figure 6F). *HIS3.3* and *SAMDC* (0.47) were the most stable genes in wt Col-0 vs. *jagger* samples (Figure 6G), while *YLS8*, *GAPC-2* and *HIS3.3* (0.37) the most stable genes in wt No-0 vs. *agp6 agp11* samples (Figure 6H). *GAPC-2* (0.73) was the least stable gene in wt Col-0 vs. *jagger* subset (Figure 6G), while *SAMDC* (0.59) was the most unstable gene in wt No-0 vs. *agp6 agp11* subset (Figure 6H).

### 3.8. Comprehensive Ranking of the Candidate Reference Genes

Finally, the RefFinder software also performs a comprehensive gene stability analysis by ranking the reference genes according to the geometric mean of the ranking values obtained for each of the four different programmes (geNorm, NormFinder, BestKeeper and ΔCt) (Figure 7).

According to the analysis, *RCE1* (1.50), *SAC52* (3.13) and *TUA2* (3.46) were the most stable genes in the flower development set, while *SAMDC* (10.00) was the most unstable gene (Figure 7A). *YLS8* (1.50), *RCE1* (2.91) and *PP2AA3* (3.46) were the top 3 ranked genes in the flower (st 1–16) subset (Figure 7B), while *TUA2* (1.86), *PP2AA3* (3.31) and *HIS3.3* (1.86) were the most stable genes in silique (st 17) samples (Figure 7C). As for pistil samples, the best option of reference genes was *RCE1* (2.11) and *GAPC-2* (2.55) (Figure 7D), while *UBC9* (1.32) and *RCE1* (2.78) were the most stable genes in anther (st 12) subgroup (Figure 7E). *TUA2* (7.04) was the least stable gene in the flower (st 1–16) samples (Figure 7B), and *GAPC-2* (10) was more unstable in silique (st 17) subgroup (Figure 7C). *ACT7* (9.74) was the most unstable gene in pistil (st 12) samples (Figure 7D), while *HIS3.3* (9.46) was the least stable gene in anther (st 12) subset (Figure 7E).

Among the genotype set, *YLS8* (1.73), *HIS3.3* (2.45) and *ACT7* (2.91) were the most stable genes, while *GAPC-2* (9.15) was the most unstable gene (Figure 7F). *HIS3.3* (1.68) and *SAMDC* (2.30) were the most stable genes in wt Col-0 vs. *jagger* samples (Figure 7G), while *YLS8* (1.00) and *GAPC-2* (2.21) were the most stable genes in wt No-0 vs. *agp6 agp11* samples (Figure 7H). *SAC52* (9.24) was the least stable gene in wt Col-0 vs. *jagger* subset (Figure 7G), while *SAMDC* (10.00) was the most unstable gene in wt No-0 vs. *agp6 agp11* subset (Figure 7H).

### 3.9. Validation of the Selected Reference Genes

In this study, to validate the reference genes, the relative expression levels of three target genes, *AGP6*, *AGP11* and *JAGGER* were quantified and normalised using the three most stable, the two most stable and two least stable reference genes for both experimental sets (Figure 8).

Regarding the flower development subsets, the results showed that similar to previous phenotypic and expression pattern studies, the *AGP6* and *AGP11* gene expression level was higher in samples containing male structures (Figure 8A and B, respectively). Concomitantly, the *JAGGER* expression level was higher on samples which included female structures (Figure 8C). The analysis showed that when the three most stable genes (*RCE1*, *SAC52* and *TUA2*) were employed for *AGP6*, *AGP11* and *JAGGER* normalisation, statistically significant differences were detected on the expression pattern of at least one of the subsets when compared to the normalisation with the two least stable reference genes (*HIS3.3* and *SAMDC*) (Figure 8A, B and C, respectively). Moreover, the use of two most stable genes (*RCE1*, *SAC52*) for *AGP6* normalisation on anthers (st 12) presented statistically significant differences to the use of three most stable genes (Figure 8A). These findings suggest that the use of reliable reference genes is important for an accurate normalisation of target genes for reproductive tissues. Regarding the expression of AGPs in the respective mutant backgrounds, as foreseeable, *AGP6* and *AGP11* were downregulated in *agp6 agp11* mutants compared to wt No-0 (Figure 8D and 8E, respectively). Likewise, the expression of *JAGGER* was downregulated in the *jagger* mutants (Figure 8F). However, no differences were detected when the three different gene combinations were used for normalisation, indicating that any of these genes may be used for normalisation in expression analysis studies on the genotype set.

## 4. Discussion

In gene expression studies, qPCR remains the most reliable method to quantify the abundance of a target gene. A key step in this technique is the normalisation of samples using appropriate reference genes as internal controls to correct the variability of RNA extraction and reverse-transcription yield and efficiency of amplification, thus allowing the comparison between different samples leading to accurate conclusions [16]. Reference genes are genes stably expressed in all samples in a given study, unaffected by experimental factors and with the same kinetics as the target genes during a qPCR [16,19,20]. Traditionally, reference genes are metabolic genes, involved in processes essential for the survival of cells such as carbon metabolism, cellular structure maintenance and protein translation, that are ubiquitously expressed in a stable and nonregulated constant level, such as actin, tubulin, GAPDH, cyclophilin, elongation factor 1α or ubiquitin [15,58,59]. However, previous studies, with both animal and plant organisms, evidenced that traditional reference genes may not be stably expressed between cell types and within cells under different conditions [24,25,26,28,59,60,61,62,63,64]. Therefore, it is important to select appropriate reference genes under specific experimental conditions.

The most universally accepted and appropriated method is to normalise against three or more validated reference genes [19,65]. Vandesompele [19] showed that normalisation with a single reference gene resulted in a inaccurate normalisation process with errors up to 3.0- and 6.4-fold in 25% and 10% of the results, respectively. Additionally, Guénin [66] showed that only 3.2% of 188 papers published in the three leading plant science journals had reference genes correctly validated in the same study or in a previous study whereas in the other 96.8% of publications, the genes were merely putatively stably expressed. Unfortunately, regardless of the increased awareness for the importance of the process of selection of the optimal number and validation of reference genes, nowadays, there is still a huge quantity of publications with qPCR experiments normalised using a single reference gene. A pilot experiment with representative samples should be performed to select stably expressed reference genes. Nevertheless, the expression stability of the selected reference genes must be evaluated in the final experiment [65,67]. In fact, the publication of the Minimum Information for Publication of Quantitative Real-Time PCR Experiments (MIQE) guidelines was an attempt to guarantee and encourage the integrity, consistency and transparency of qPCR experiments, leading to more reliable, comparable, and unambiguous results [16]. However, Bustin [68] even the enrolled two surveys covering 1700 publications with qPCR experiments that still exposed a lack of transparency on reporting essential technical and quality-control information, leading to inaccurate biological results.

In the plant reproduction field, qPCR is a widely performed technique, using templates for biological samples from different flower structures and/or developmental stages. However, there is still a literature gap concerning reference gene validation for these types of biological samples. We are aware that different reproductive structures involve specific developmental processes, which, in turn, require dynamical expression patterns [69,70,71]. These distinct expression profiles elevate the importance of choosing and validating the reference genes that better suit the biological groups under analysis. To our knowledge, in the *A. thaliana*, refs. [27,60,63,72] are the only studies to appropriate validate candidate reference genes for qPCR analysis in heat stress, various conditions (such as developmental series, shoot and root abiotic stress series, biotic stress series, photomorphogenic light series, and hormone series), metal exposure and seed samples.

In the present study, ten candidate reference genes (*UBC9*, *ACT7*, *GAPC-2*, *RCE1*, *PP2AA3*, *TUA2*, *SAC52*, *YLS8*, *SAMDC* and *HIS3.3*) were selected for fu rther validation in several biological samples containing different flower structures and stages and for a comparison between genotypes. To the best of our knowledge, this is the first analysis to select and validate suitable reference genes for expression analysis on reproductive tissues of *A. thaliana*. For the ranking of the best reference genes, four statistical algorithms (geNorm, NormFinder, BestKeeper and ΔCt method) were used. Since the output information between the different software may not be consistent, we used RefFinder, which combines the four algorithms and gives a final output of the more stable reference genes [29].

According to RefFinder, which combines the four different software outcomes, the three most stable genes found were *RCE1*, *SAC52* and *TUA2* for flower development and *YLS8*, *HIS3.3* and *ACT7* for genotype set. It is worth noting that these two sets encompassed different developmental stages and/or tissues, showing a higher heterogeneity when compared to the subsets. Therefore, the most stable gene ranking of each subset is slightly different from the correspondent set: *YLS8*, *RCE1,* and *PP2AA3* for flower (st 1–16) subset; *TUA2*, *PP2AA3,* and *HIS3.3* for silique (st 17) subset; *RCE1*, *GAPC-2* and *TUA2* for pistil (st 12) samples; *UBC9*, *RCE1,* and *SAMDC* for anther (st 12) subset; *HIS3.3*, *SAMDC,* and *YLS8* for wt Col-0 vs. *jagger* samples and finally, *YLS8*, *GAPC-2,* and *HIS3.3* for wt No-0 vs. *agp6 agp11* samples. From this analysis we can also inferred if a gene is stable in a specific organ or across a more variable group. The top-ranking genes for the flower development set and flower (st 1–16) subset would be more stably expressed across different tissues and developmental stages compared to the ones that only appeared when a specific organ, such as silique (st 17), pistil (st 12), or anther (st 12) was used. From the genotype set results, we can conclude that the most stable genes are not being affected by the absence of the three *AGP* genes. When several algorithms are employed in a stability analysis, the output can significantly vary due to different assumptions and computations. An integration of the information for different rankings may be difficult, as each software has its strengths, weaknesses, and suitable application conditions [73]. This may be done with complementary tools that merge all the results of different algorithms into a comprehensive ranking, such as RefFinder [29]. This tool calculates the geometric mean of the ranking values of the four programmes for one final overall ranking [29]. In RefFinder, the input is the raw Cq values, assuming for all genes a 100% PCR efficiency. Since the PCR efficiencies for each gene are not taken into account, the RefFinder outputs may be biased and have an impact on the expression stability ranking of the reference genes. It is the user’s responsibility to assure that the primer design is correct in order to obtain efficiencies that vary no more than 10% from the optimal 100% efficiency [73], as the ones we obtained in the present study. 

In our study case, the differences between the underlying algorithms were minimal since both geNorm, NormFinder, and ΔCt identified at least the same gene as being the most stable one for each analysed situation. Only the ranking given by BestKeeper presented a different output, for instance, this algorithm identified *UBC9* as an appropriate reference gene for flower (st 1–16) subset, while this gene was identified on the other software as one of the least stable ones. Previous studies have reported such disparities between BestKeeper and the other algorithms [74,75,76,77] and the reason might be associated with the algorithm configuration itself.

GeNorm algorithm is also able to define an optimal number of reference genes to be used for qPCR normalisation by calculating a pairwise variation value [19]. For all the analysis conducted, the V_2_/V_3_ were below the acceptable threshold value of 0.15, meaning that two of the ten candidate genes would be sufficient to accurately normalise our samples. This result was expected and shows the effectiveness of our experimental design, specifically in trying to avoid as much external variation that could affect the experiment, such as RNA/cDNA quality and concentration, primer design, and efficiency. Hence, the Cq values of the ten candidate genes did not show a wide variation whereby geNorm revealed the necessity of only two reference genes. Nevertheless, a visual interpretation of the pairwise variations can also be informative. In our case, the analysis suggests that both sets and subsets would benefit from using three reference genes for normalisation, since the V_3_/V_4_ is smaller than V_2_/V_3_.

Finally, in order to validate the selected reference genes for each situation, the expression levels of *AGP6*, *AGP11* and *JAGGER* were analysed. These three genes were selected due to their expression patterns in the reproductive tissues: *AGP6* and *AGP11,* which are present mainly in the male tissues [11,12,13], while *JAGGER* shows a higher expression in the female tissues [14]. Indeed, our results from the flower development set are in accordance with these expression patterns. However, the statistical analysis detected significant differences when different combinations of reference genes were applied to the tissues where *AGP6*, *AGP11* and *JAGGER* are mostly expressed. These findings corroborate with the importance of selecting a reliable number of reference genes, particularly when different types of tissues are under the same gene expression analysis. The differences between the genotypes were less sensitive to the choice of the set of reference genes than the tissue-specific differences. The fact that choosing the two least stably expressed genes as reference genes yielded virtually identical results as choosing the three most stably expressed genes means that any combination of at least two of our ten candidate genes may be safely used as reference genes for comparisons across genotypes.

## 5. Conclusions

We have evaluated ten candidate reference genes to normalise gene expression using qPCR in different reproductive tissues in *A. thaliana*. The findings of this study, using geNorm, NormFinder, BestKeeper and ΔCt method on RefFinder tool, indicate that *RCE1*, *SAC52* and *TUA2* are appropriate reference genes for comparing gene expression across a set of flower development samples, whilst *YLS8*, *HIS3.3* and *ACT7* are a proper combination of reference genes for a genotype situation. For the first time, a study was conducted to assess the stability of candidate reference genes in different reproductive tissues in *A. thaliana*, providing a framework that will contribute to the consistency and accuracy of transcripts quantification. Indeed, these principles extend to any field of work and to any other candidates for reference genes. Furthermore, our results will enable researchers to save time and costs when performing gene expression studies with qPCR. Nevertheless, we recommend the re-validation of the reference genes whenever a new study is performed.

## Figures and Tables

**Figure 1 biomolecules-13-00463-f001:**
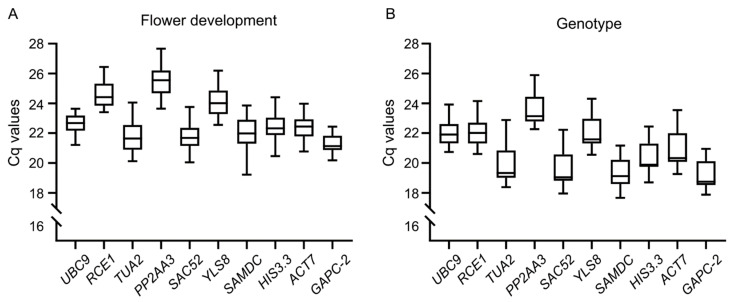
Distribution of Cq values obtained from the ten candidate reference genes. (**A**) Cq values of the ten candidate reference genes in the flower development group which included 21 samples corresponding to flower (st 1–16), silique (st 17), pistil (st 12), and anther (st 12) samples. (**B**) Cq values of ten candidate reference genes in the genotype set that comprised 12 samples corresponding to wt Col-0, *jagger*, wt No-0 and *agp6 agp11* samples. The solid line within each box represents the median Cq values and the boxes denote the 1st and 3rd quartiles.

**Figure 2 biomolecules-13-00463-f002:**
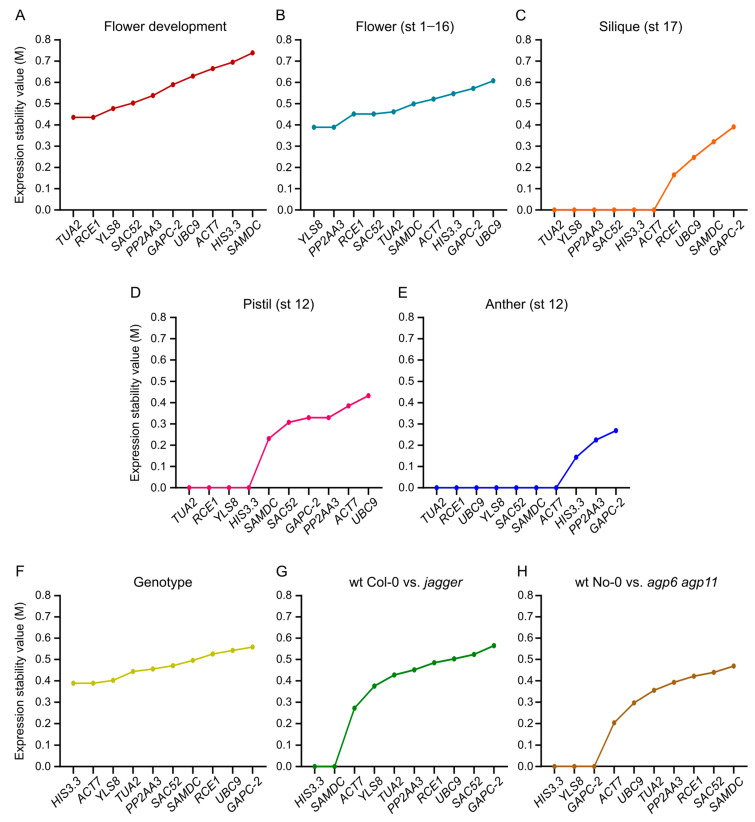
Expression stability values (M) of the ten candidate reference genes calculated by geNorm in the RefFinder software. (**A**) M values of the ten candidate reference genes in the flower development group which included 21 samples corresponding to flower (st 1–16), silique (st 17), pistil (st 12), and anther (st 12) samples. (**B**) M values of the ten candidate reference genes in 12 flower (st 1–16) samples which included flower st 1–6, flower st 7–12, flower st 13–14, and flower st 15–16 samples. (**C**) M values of the ten candidate reference genes in the three silique (st 17) samples. (**D**) M values of the ten candidate reference genes in the three pistil (st 12) samples. (**E**) M values of the ten candidate reference genes in the three anther (st 12) samples. (**F**) M values of the ten candidate reference genes in the genotype set that comprised 12 samples corresponding to wt Col-0, *jagger*, wt No-0 and *agp6 agp11* samples. (**G**) M values of the ten candidate reference genes in six wt Col-0 vs. *jagger* samples that included three wt Col-0 and three *jagger* samples. (**H**) M values of the ten candidate reference genes in six wt No-0 vs. *agp6 agp11* samples which included three wt No-0 and three *agp6 agp11* samples. The genes are arranged from the most to the least stable gene (from left to right). Abbreviations: Col-0, Columbia-0; No-0, Nossen-0; st, stage; wt, wild-type.

**Figure 3 biomolecules-13-00463-f003:**
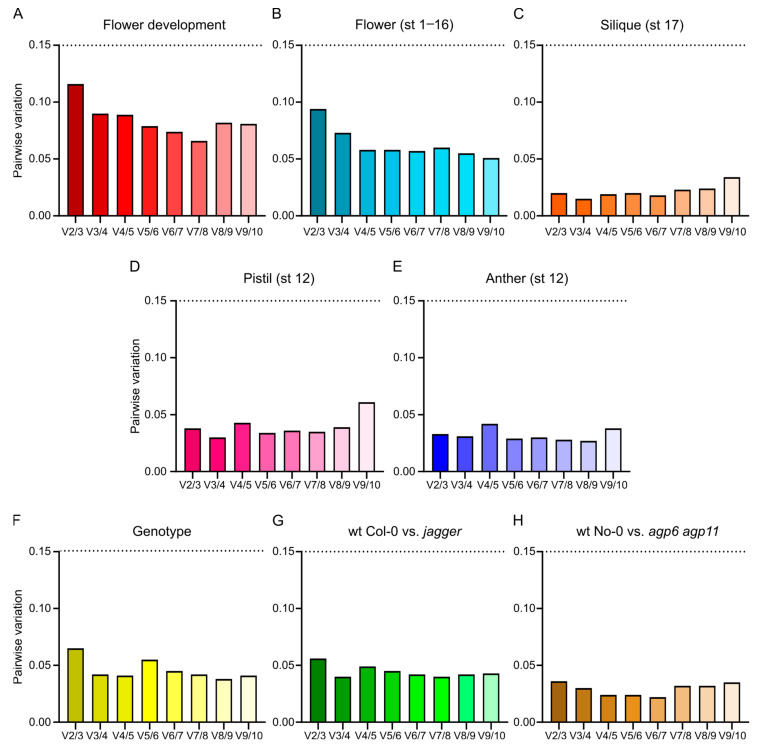
Pairwise variation (V_n_/V_n+1_) of the ten candidate reference genes using geNorm. (**A**) Pairwise variation values of the ten candidate reference genes in the flower development group which included 21 samples corresponding to flower (st 1–16), silique (st 17), pistil (st 12), and anther (st 12) samples. (**B**) Pairwise variation values of the ten candidate reference genes in 12 flower (st 1–16) samples which included flower st 1–6, flower st 7–12, flower st 13–14, and flower st 15–16 samples. (**C**) Pairwise variation values of the ten candidate reference genes in the three silique (st 17) samples. (**D**) Pairwise variation values of the ten candidate reference genes in the three pistil (st 12) samples. (**E**) Pairwise variation values of the ten candidate reference genes in the three anther (st 12) samples. (**F**) Pairwise variation values of the ten candidate reference genes in the genotype set that comprised 12 samples corresponding to wt Col-0, *jagger*, wt No-0 and *agp6 agp11* samples. (**G**) Pairwise variation values of the ten candidate reference genes in six wt Col-0 vs. *jagger* samples that included three wt Col-0 and three *jagger* samples. (**H**) Pairwise variation values of the ten candidate reference genes in six wt No-0 vs. *agp6 agp11* samples which included three wt No-0 and three *agp6 agp11* samples. The pointed line represents the cut-off value of 0.15. Abbreviations: Col-0, Columbia-0; No-0, Nossen-0; st, stage; wt, wild-type.

**Figure 4 biomolecules-13-00463-f004:**
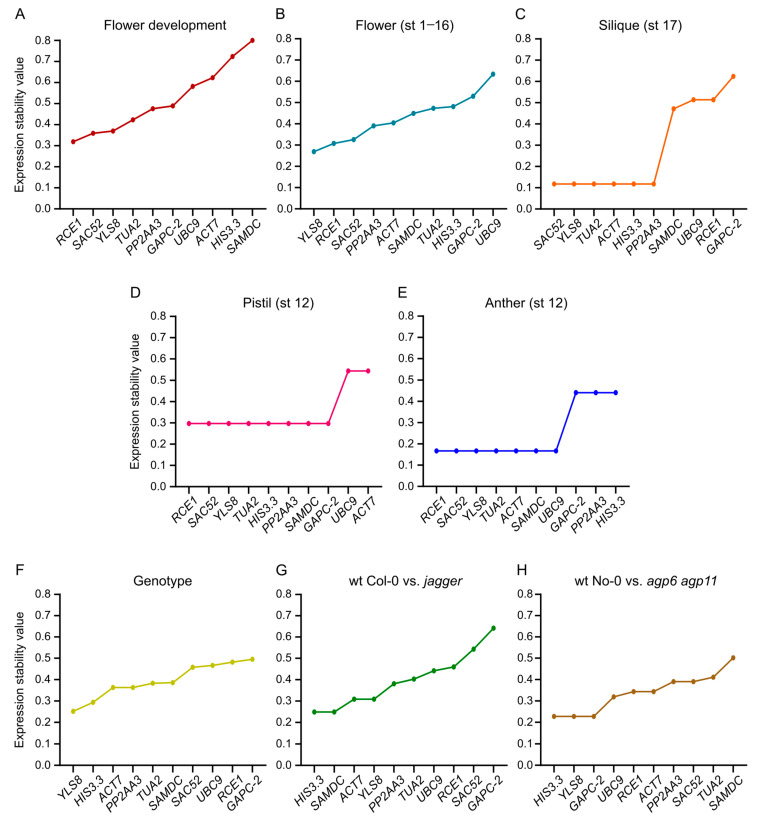
Expression stability values of the ten candidate reference genes calculated by NormFinder in the RefFinder software. (**A**) Expression stability values of the ten candidate reference genes in the flower development group which included 21 samples corresponding to flower (st 1–16), silique (st 17), pistil (st 12), and anther (st 12) samples. (**B**) Expression stability values of the ten candidate reference genes in 12 flower (st 1–16) samples which included flower st 1–6, flower st 7–12, flower st 13–14, and flower st 15–16 samples. (**C**) Expression stability values of the ten candidate reference genes in the three silique (st 17) samples. (**D**) Expression stability values of the ten candidate reference genes in the three pistil (st 12) samples. (**E**) Expression stability values of the ten candidate reference genes in the three anther (st 12) samples. (**F**) Expression stability values of the ten candidate reference genes in the genotype set that comprised 12 samples corresponding to wt Col-0, *jagger*, wt No-0 and *agp6 agp11* samples. (**G**) Expression stability values of the ten candidate reference genes in six wt Col-0 vs. *jagger* samples that included three wt Col-0 and three *jagger* samples. (**H**) Expression stability values of the ten candidate reference genes in six wt No-0 vs. *agp6 agp11* samples which included three wt No-0 and three *agp6 agp11* samples. The genes are arranged from the most to the least stable gene (from left to right). Abbreviations: Col-0, Columbia-0; No-0, Nossen-0; st, stage; wt, wild-type.

**Figure 5 biomolecules-13-00463-f005:**
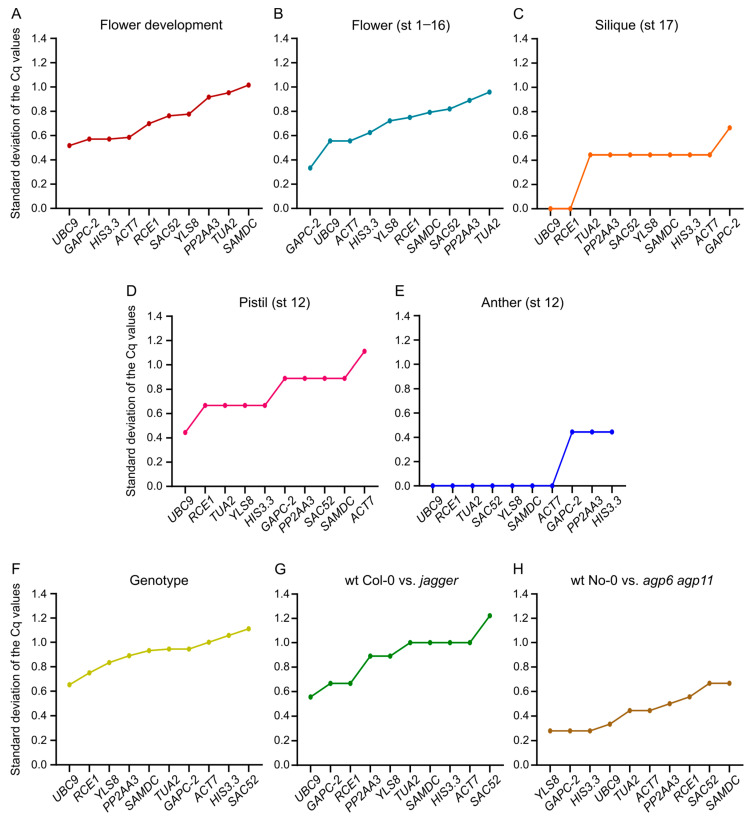
Standard deviation (SD) of the Cq values of ten candidate reference genes calculated by BestKeeper in the RefFinder software. (**A**) SD values of the ten candidate reference genes in the flower development group which included 21 samples corresponding to flower (st 1–16), silique (st 17), pistil (st 12), and anther (st 12) samples. (**B**) SD values of the ten candidate reference genes in 12 flower (st 1–16) samples which included flower st 1–6, flower st 7–12, flower st 13–14, and flower st 15–16 samples. (**C**) SD values of the ten candidate reference genes in the three silique (st 17) samples. (**D**) SD values of the ten candidate reference genes in the three pistil (st 12) samples. (**E**) SD values of the ten candidate reference genes in the three anther (st 12) samples. (**F**) SD values of the ten candidate reference genes in the genotype set that comprised 12 samples corresponding to wt Col-0, *jagger*, wt No-0 and *agp6 agp11* samples. (**G**) SD values of the ten candidate reference genes in six wt Col-0 vs. *jagger* samples that included three wt Col-0 and three *jagger* samples. (**H**) SD values of ten candidate reference genes in six wt No-0 vs. *agp6 agp11* samples which included three wt No-0 and three *agp6 agp11* samples. The genes are arranged from the most to the least stable gene (from left to right). Abbreviations: Col-0, Columbia-0; No-0, Nossen-0; st, stage; wt, wild-type.

**Figure 6 biomolecules-13-00463-f006:**
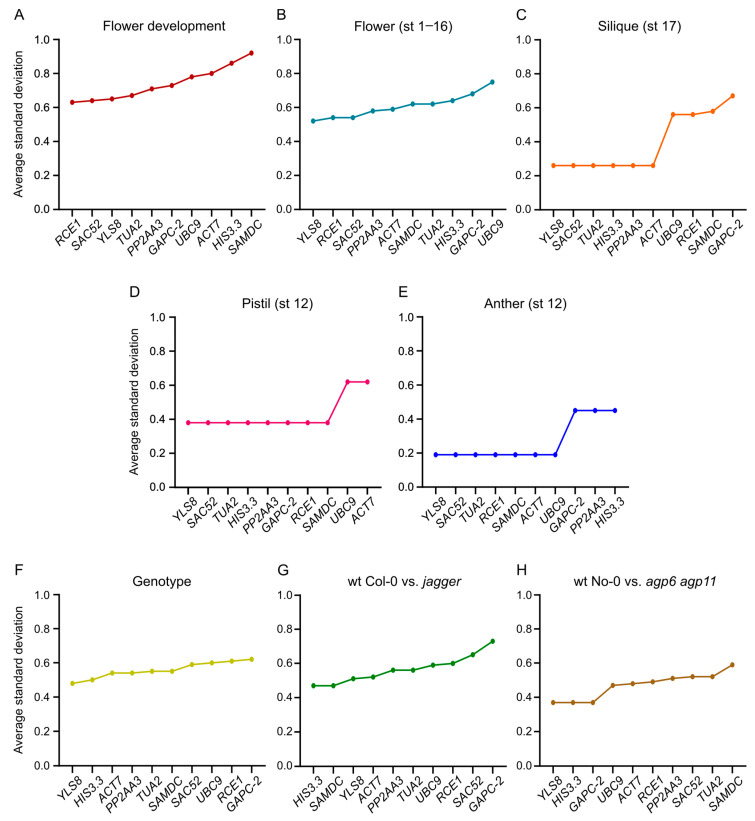
Average standard deviation (SD) of the ten candidate reference genes calculated by the ΔCt in the RefFinder software. (**A**) SD values of the ten candidate reference genes in the flower development group which included 21 samples corresponding to flower (st 1–16), silique (st 17), pistil (st 12), and anther (st 12) samples. (**B**) SD values of the ten candidate reference genes in 12 flower (st 1–16) samples which included flower st 1–6, flower st 7–12, flower st 13–14, and flower st 15–16 samples. (**C**) SD values of the ten candidate reference genes in the three silique (st 17) samples. (**D**) SD values of the ten candidate reference genes in the three pistil (st 12) samples. (**E**) SD values of the ten candidate reference genes in the three anther (st 12) samples. (**F**) SD values of the ten candidate reference genes in the genotype set that comprised 12 samples corresponding to wt Col-0, *jagger*, wt No-0 and *agp6 agp11* samples. (**G**) SD values of the ten candidate reference genes in six wt Col-0 vs. *jagger* samples that included three wt Col-0 and three *jagger* samples. (**H**) SD values of ten candidate reference genes in six wt No-0 vs. *agp6 agp11* samples which included three wt No-0 and three *agp6 agp11* samples. The genes are arranged from the most to the least stable gene (from left to right). Abbreviations: Col-0, Columbia-0; No-0, Nossen-0; st, stage; wt, wild-type.

**Figure 7 biomolecules-13-00463-f007:**
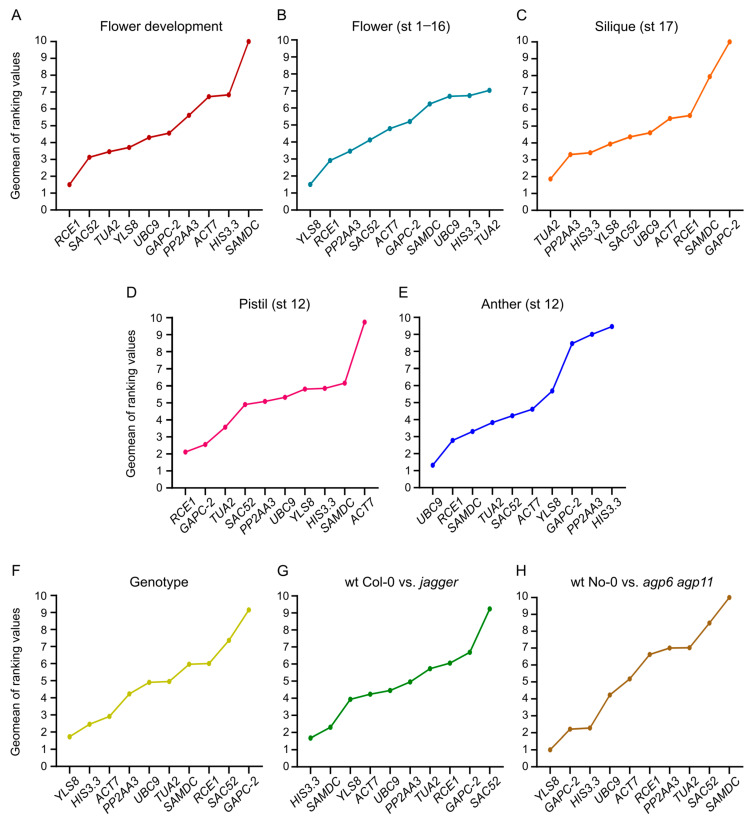
Comprehensive ranking values of the candidate reference genes obtained from geNorm, NormFinder, BestKeeper and ΔCt. (**A**) Ranking values of the ten candidate reference genes in the flower development group which included 21 samples corresponding to flower (st 1–16), silique (st 17), pistil (st 12), and anther (st 12) samples. (**B**) Ranking values of the ten candidate reference genes in 12 flower (st 1–16) samples which included flower st 1-6, flower st 7–12, flower st 13–14 and flower st 15–16 samples. (**C**) Ranking values of the ten candidate reference genes in the three silique (st 17) samples. (**D**) Ranking values of the ten candidate reference genes in thethree pistil (st 12) samples. (**E**) Ranking values of the ten candidate reference genes in the three anther (st 12) samples. (**F**) Ranking values of ten candidate reference genes in the genotype set that comprised 12 samples corresponding to wt Col-0, *jagger*, wt No-0 and *agp6 agp11* samples. (**G**) Ranking values of the ten candidate reference genes in six wt Col-0 vs. *jagger* samples that included three wt Col-0 and three *jagger* samples. (**H**) Ranking values of the ten candidate reference genes in six wt No-0 vs. *agp6 agp11* samples which included three wt No-0 and three *agp6 agp11* samples. The genes are arranged from the most to the least stable gene (from left to right). Abbreviations: Col-0, Columbia-0; No-0, Nossen-0; st, stage; wt, wild-type.

**Figure 8 biomolecules-13-00463-f008:**
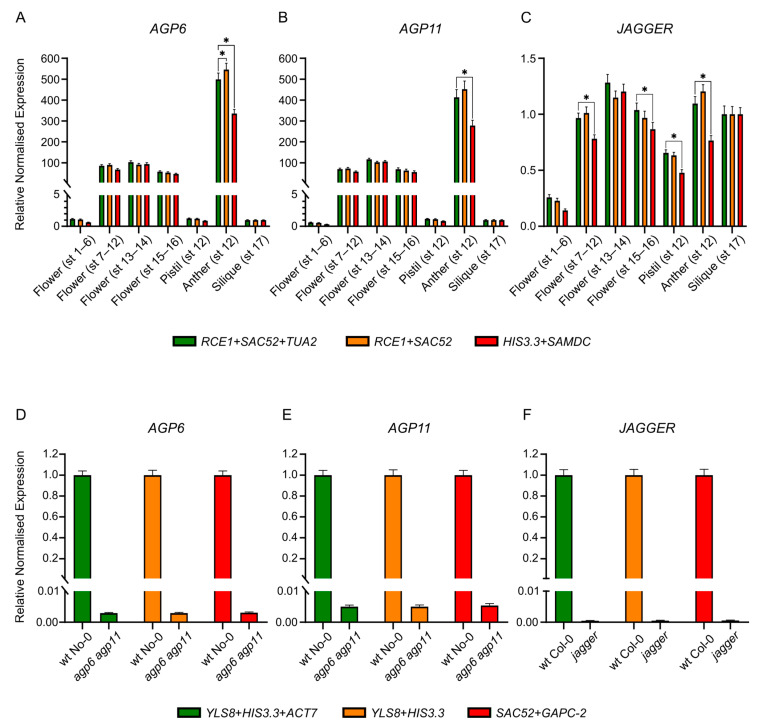
Relative expression levels of *AGP6*, *AGP11* and *JAGGER* normalised by the three most stable reference genes (green bars), the two most stable reference genes (orange bars) and the two least stable reference genes (red bars) combinations. Relative expression levels of *AGP6* (**A**), *AGP11* (**B**) and *JAGGER* (**C**) in the flower development subsets, namely flower (st 1–6), flower (st 7–12), flower (st 13–14), flower (st 15–16), anther (st 12), pistil (st 12) and silique (st 17). The data correspond to the ratio of the expression compared with the silique (st 17). Relative expression levels of *AGP6* (**D**) and *AGP11* (**E**) in the wt No-0 vs. *agp6 agp11* subset and *JAGGER* (**F**) in the wt Col-0 vs. *jagger* subset. The data correspond to the ratio of the expression compared with the wt No-0 for *AGP6* and *AGP11* or wt Col-0 for *JAGGER*. Error bars represent the standard error of the mean of three independent biological replicates, each with three technical replicates. Statistical analyses were performed using a two-way ANOVA followed by Dunnett’s multiple comparisons test. Asterisks represent statistically significant differences (α < 0.05) between the three different combinations of reference genes in each sample. Abbreviations: Col-0, Columbia-0; No-0, Nossen-0; st, stage; wt, wild-type.

**Table 1 biomolecules-13-00463-t001:** List of candidate reference genes and primer sequences used in the qPCR analysis.

Locus	Gene Symbol	Gene Name	Primer Sequences Forward and Reverse (5′–3′)	Amplicon Length (bp)
*AT4G27960*	*UBC9*	*UBIQUITIN CONJUGATING ENZYME 9*	AGATGATCCTTTGGTCCCTGAG	114
			CAGTATTTGTGTCAGCCCATGG	
*AT5G09810*	*ACT7*	*ACTIN 7*	ATCAATCCTTGCATCCCTCAGC	72
			GGACCTGACTCATCGTACTCAC	
*AT1G13440*	*GAPC-2*	*GLYCERALDEHYDE-3-PHOSPHATE DEHYDROGENASE C2*	TGGGGTTACAGTTCTCGTGTC	83
			ACCACACACAAACTCTCGCC	
*AT4G36800*	*RCE1*	*RUB1 CONJUGATING ENZYME 1*	CGGTGGATATGTCGGTCAG	135
			AACGAGGGTCCTTGAGAAAGAG	
*AT1G50010*	*TUA2*	*TUBULIN ALPHA-2 CHAIN*	CATTGAGAGACCCACCTACACC	78
			AACCTCAGAGAAGCAGTCAAGG	
*AT1G13320*	*PP2AA3*	*PROTEIN PHOSPHATASE 2A SUBUNIT A3*	TGTTCCAAACTCTTACCTGCGG	136
			ATGGCCGTATCATGTTCTCCAC	
*AT1G14320*	*SAC52*	*SUPPRESSOR OF ACAULIS 52*	CGTCGTGCTAAGTTCAAGTTCC	108
			CTTCTCTTGCCTCAACTTGGTG	
*AT5G08290*	*YLS8*	*YELLOW-LEAF-SPECIFIC GENE 8*	AAGATCAACTGGGCTCTCAAGG	141
			TGGGAAGCTCGATTAGTAACGG	
*AT3G02470*	*SAMDC*	*S-ADENOSYLMETHIONINE DECARBOXYLASE*	TTGGTAAGTACTGTGGATCGCC	101
			CTGCTAGATTCCCTCGTCCTTC	
*AT4G40030*	*HIS3.3*	*HISTONE 3.3*	ACCTTTGTGCCATTCATGCC	78
			GTTCACCTCTGATACGACGAGC	

**Table 2 biomolecules-13-00463-t002:** Amplification efficiencies, correlation coefficients (R^2^), slope and melting temperature of qPCR primers of ten candidate reference genes.

Gene Symbol	Efficiency (%)	R^2^	Slope	Melting Temperature (°C)
*UBC9*	91.2	1	−3.553	82.5
*ACT7*	105.6	0.997	−3.194	79
*GAPC-2*	93.7	0.999	−3.483	81.5
*RCE1*	96.6	0.999	−3.406	81
*TUA2*	104.4	0.999	−3.221	80.5
*PP2AA3*	100.3	0.998	−3.314	81.5
*SAC52*	96.8	0.999	−3.402	81.5
*YLS8*	90	0.996	−3.587	82.5
*SAMDC*	107.5	0.989	−3.155	82.5
*HIS3.3*	101.2	0.992	−3.293	81.5

## Data Availability

All data are contained within the manuscript and Appendix A.

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
