# Peer review of "I Choose You: Selecting Accurate Reference Genes for qPCR Expression Analysis in Reproductive Tissues in Arabidopsis thaliana"

_biomolecules, 2023, doi:10.3390/biom13030463_

Round 1

Reviewer 1 Report (New Reviewer)

In this work, entitled « I choose you: selecting accurate reference genes for qPCR expression analysis in reproductive tissues in Arabidopsis thaliana », Ferreira et al analyzed the expression of 10 endogenous control genes for qPCR analysis of gene expression in reproductive tissue. The paper is well written, and the data are sound and useful for the community.

Minor comments:

·      It would be nice to include in supplements, snapshots of the expression of those genes in publicly available RNA seq such as Klepikova et al 2016

·      Please rephrase the following sentences for clarity

o   Parallelly, JAGGER, an AGP expressed solely in the female tissues/cells, was described as essential in the signalling pathway responsible for blockage of PT attraction once fertilisation is achieved.

o   The study of plant reproduction is a necessary, crucial and much relevant activity as it will provide knowledge and tools for maintaining successful fertilisation under environmental stress conditions and for the production of resilient seeds.

o   However, this assumption was shown to be wrong either for animal or plant organisms

Author Response

Reviewer 2 Report (New Reviewer)

The manuscript really described excellent analyses for reference gene choices in qPCR. These results could provide resources for proper control genes in qPCR in reproductive biology. The manuscript is well-written. I believe the manuscript will attract general interest in biologists as qPCR is a quite common and highly frequent experiment now.

I only have a minor comment:

The double mutant "agp6agp11" should be "agp6 agp11". It should have a space between two genes. Please check and correct them throughout the manuscript.

Author Response

Reviewer 3 Report (New Reviewer)

I was initially very interested in reading this manuscript based on the title and the abstract. This is a potentially important topic. Unfortunately, after reading it I am rather disappointed.

As the abstract notes, “a considerable number of publications continues to present qPCR results normalised to a single reference gene and, to our knowledge, no evaluation has been carried in specific reproductive tissues of Arabidopsis thaliana”.

The problem is that this study does not really do that either! The analyses focus on analyzing the expression of eight potential qPCR reference genes in:

1. Mature inflorescences of two ecotypes of A. thaliana (Col-0) and No-0) and two mutants in the Col-0 background.

2. A developmental profile of whole inflorescences using stages 1-6 (combined), 7 to 12 (combined), 13-14 (combined) and 15-16 (combined), all from Col-0.

3. Pistils and anthers from stage 12 flowers; wt Col-0 siliques stage 17.

General Comments:

i.  Though largely well crafted, the manuscript could be improved with English editing (grammar issues). As far as I can tell the analyses performed are appropriate and have been done well – but I am not familiar with most of the software packages used.

ii. It would help readers of this manuscript to include an explanation of the strategies used to assess the genes being studied. One point of using reference house-keeping genes is to account for differences in samples mRNA concentrations, pipetting errors, etc, all of which can be considerable. Trying to assess which house-keeping genes are best is obviously challenging if you cannot use one of them to equalize the samples! The strategy being used here – comparing various measures of stability of expression etc, seem to be a good solution to the problem, but it would help the reader if the rationale was clearly explained.

iii. My main problem with this manuscript is that the samples in set 3 are the only samples that could be described as “specific reproductive tissues”! For me this is problematic given the complexity of organs and tissues in floral buds, which contain developing male and female gametophytes, as well as maternal sporophytic tissues. I disagree that inflorescences are “the most commonly used sample type in plant reproduction when confirming the expression of a mutated gene”. Researchers are generally focused on either male or female organs. In my view, this analysis would be considerably more useful if male and female organs had been assessed separately through the developmental profile. This does not invalidate the results presented, and they do have some interesting implications, but it does reduce the significance of the manuscript – particularly as this analysis provides information that is largely of value only to Arabidopsis researchers. This is clearly a decision for the editor, but for me this reduces the potential readership, and hence impact, of this manuscript below that needed for publication in Biomolecules (this may not be the case for a lower impact plant specific journal). I am recommending rejection because for me to warrant publication in Biomolecules this work needs to include developmental profiles of anthers and pistils. According to the reviewer guidelines suggesting "additional experiments" should result in a rejection recommendation).

Author Response

Reviewer 4 Report (New Reviewer)

Dear Authors,

I have reviewed your manuscript "I Choose You: Selecting Accurate Reference Genes for qPCR Expression Analysis in Reproductive Tissues in Arabidopsis thaliana", submitted for publication in Biomolecules.

First of all I want to congratulate you on your excellently performed research and well-written manuscript. Your work has a high value which goes a long way beyond its own results, since it demonstrates in detail an example of good practice on how the choice of reference genes should be carried out for gene expression studies, even outside of the context of reproductive development which is the primary scope of your paper.

Nevertheless, I have minor suggestions for improvement, regarding, above all, more detailed clarification of a number of points throughout the manuscript, as well as minor language corrections. Please note that I do not consider any of these to be absolutely necessary for the publication of your work; they are just intended to make it more thoroughly explained. My suggestions are as follows:

·         Abstract:

o   line 15-16: please revise for additional clarity: "no comparative evaluation of multiple reference genes has been carried out in specific reproductive tissues of Arabidopsis thaliana"

o   line 17: an additional "A" is missing from "PP2AA3"

·         Language:

o   Your manuscript is written in excellent English, but I do have one important remark. Please avoid the use of the Saxon genitive at points such as in lines 13, 41, 242, 279, 295, 627 ("mutants backgrounds", "seeds development", "samples collection", "amplicons length", "values distribution", "gene's ranking"). Please remember that you are doing research in plant development, not "plants development", "plant's development", nor "plants' development" - even though your research is not focused on a single plant but on all of them. Accordingly, you should say "in mutant backgrounds", "seed development", "sample collection", "amplicon lengths", "value distribution", and "gene ranking".

·         Introduction:

o   line 32: you may delete "A. thaliana" which is in the brackets. It is widely accepted to use the abbreviated form of a species name without further explanation after you have once spelled out its full name.

o   line 55: please revise this sentence to make it less dramatic. I suggest something like "The study of plant reproduction is very important as it will provide..."

o   line 67-68: please add a clarification: "minimizing the artefactual variability of total cDNA abundance among samples" or something like that. The same idea is presented in a more appropriate way within the Discussion, in the lines 657-658; but, just saying "minimizing the variability among samples" is not sufficient to express the idea that you had here.

o   line 73-74: "for both animal and plant organisms" (not "either/or", but "both/and")

·         Material & Methods:

o   Section 2.2, first paragraph. This paragraph could benefit a lot from several clarifications:

§  line 105: A brief summarisation of the criteria for the literature search would be welcome here, if possible.

§  line 107: Here as well, a brief elaboration of how the ten candidate genes were chosen, could be provided, if possible.

§  line 108: Please provide a brief explanation of what exactly is meant by "male". (Please provide the same explanation within the Supplementary Material as well.)

o   Section 2.4, line 200: "WERE analysed"

·         Results:

o   Section 3.1: The text in this section is a repetition of the text in 2.3, although here it is written more comprehensively. I recommend to remove the existing text from 2.3 and replace it with the text that is currently in 3.1. If the current 2.3 contains any details that are not in the current 3.1, then make sure to add those to the revised version of 2.3. However, none of these belong to the Results section.

o   Section 3.2: A concluding sentence, summarising the results presented in Table S1, should be added at the end of the section 3.2.

o   line 300-302: Please double-check the accuracy of this statement, it does not seem to completely correspond with the values in the graphs in Figure 1.

o   line 306: Please replace "either" with "both".

o   Section 3.10: At multiple points (e.g., lines 493, 534, 556) you refer to the two least stable genes as the two "less stable" genes. Please revise.

·         Figures:

o   Figure 2 needs a whole lot of clarifications:

§  First of all, since it shows 8 diagrams, I recommend to label it as consisting of eight subfigures (A-H) instead of just two.

§  Next: What is it exactly, that is compared in each graph? Reading your whole paper, I can guess the answer to this question, but it still needs to be fully spelled out. Only the last two subfigures (G and H) are close to being clear about it. For instance, in Figure 2G, wt-Col0 is compared to the jagger mutant. What does that exactly mean? Is the stability of the analysed genes tested across all the 6 samples as 6 separate entities (three wt-Col0 inflorescence samples and three jagger inflorescence samples), or are the mean values of the genotypes compared, and how? Same goes for Figure 2H.

§  Regarding the remaining subfigures, there is even less clarity. What is compared in the "Genotype" subfigure (2F)? Is it a comparison between the 4 genotypes, or between the 12 samples corresponding to the 4 genotypes? And what is exactly compared in each of the subfigures 2A-2E? For instance, each of the subfigures 2C-2E corresponds to only one set of samples. What is being compared in them, then? And so on.

§  Please provide a thorough clarification of the choice of various comparisons shown in these figures within the text of the Results, and a thorough explanation of each of the subfigures, within the Figure caption.

o   The same set of comments applies also to the Figures 4-7.

o   Also, it would be worth noting that, if the diagrams 2C-E (and 4,5,6,7 C-E) represent only the comparisons between the three biological replicates of a single tissue type, the stability of gene expression across these samples has much less practical value than what is shown in the diagrams A and B. This should be put forward at some place within the Discussion, or even within the Results section.

·         Discussion:

o   line 582: please delete the word "extremely" from "extremely important"

o   line 603: using as templateS, in plural.

o   line 616-617: and for a comparison between genotypes

o   line 628: please correct "from the sets" to "between the sets"

o   line 641: takeN into account

o   line 674-677: The last sentence in the last paragraph of the Discussion sounds like a mere reiteration of the results rather than true discussion. Please revise. You should put forward the fact, that the differences between the genotypes were less sensitive to the choice of the set of the reference genes, than the tissue-specific differences were. The fact that choosing the two least stably expressed genes as reference genes yielded virtually identical results as choosing the three most stably expressed genes (Figure 8B), means that any combination of at least two out of your 10 candidate genes, may be safely used as reference genes for comparisons across genotypes.

·         Conclusions:

o   Although your research was performed for the validation of reference genes for the investigation of gene expression in reproductive tissues, the value of your paper is broader than that, and the principles on which your research relied can (and need to) be replicated even outside of the reproductive tissue-context, and outside of the particular reference genes that you used here. This should be more boldly stated in the Conclusions section, and, I would suggest, also in the Abstract of your paper.

·         Authorship Contribution Statement: Please re-write in accordance with the CRediT Taxonomy guidelines: https://credit.niso.org/

·         Acknowledgments: Please remove the template text from the Acknowledgments section.

I look forward to reading your published paper in Biomolecules.

Reviewer

Round 2

Reviewer 3 Report (New Reviewer)

I find the authors responses adequate.

This manuscript is a resubmission of an earlier submission. The following is a list of the peer review reports and author responses from that submission.

Round 1

Reviewer 1 Report

This study provided a pool of appropriate reference genes for expression studies in reproductive tissues of A. thaliana. However, I have several minor comments as follows:

 1. Now the keyword number is seven, so it is better to reduce the keyword number, such as Arabidopsis thaliana, Reference Genes, Reproductive tissues, Expression analysis, Normalisation

 2. In the results section 3.9, please identify the most stable genes in silique, pistil and anther samples, respectively, which is not consistent with the Figure 7A, such as “while HIS3.3 (0), ACT7 (0) and TUA2 (1.86) were the most 471 stable genes in silique (st 17) samples”, in the Fig. the most stable gene is TUR2. Please carefully check all the figures.

 3. Please check the format of the references carefully, such as references 2, 5, 23, 33, 44, 45, 46, 50, 51, 57, 58, 60 and 61, etc.

Reviewer 2 Report

In this manuscript, the authors tested ten genes for suitable interna control of qPRC analysis in flowers. The techniques they applied to analyze the data and the samples they harvested at different stages are representative with detailed analysis and solid interpretation. The authors also validated the data with three flower specific target genes and gave clear conclusion for researchers in the related area to follow, which is appreciate. Therefore, I recommend this manuscript to be accepted.

Reviewer 3 Report

The manuscript entitled “I choose you: selecting accurate reference genes for qPCR expression analysis in reproductive tissues in Arabidopsis thaliana” investigated the expression levels of ten candidate genes at developmental stages and in different genotypes for selecting a subset of reference genes for qRT-PCR analysis. Finally, some references genes are chosen for qRT-PCR analysis in reproductive tissues of Arabidopsis. Because many reports regarding selection of reference gene sets for qRT-PCR analyses in a variety of tissues, developmental stages, and under abiotic and biotic stress conditions have been published, I think that this manuscript has a lack of novelty. Thus, I could not recommend this manuscript for publication in Biomolecules journal.

Reviewer 4 Report

The work of Ferreira et al describe the identification of a set of reference genes for Arabidopsis thaliana. The claimed novelty of the work is based on the use of reproductive tissue. They use different developmental stages of inflorescences, pistils and anthers. Arabidopsis is a very well-established model system. Starting in the early 2000, there has been a very large number of trascriptomic studies using both microarrays and RNAseq from different flower development stages, pistils, anthers, under various growth conditions. Those studies have already identified reference gene genes using complete transcriptomics, which is by far a much better option, if qPCRs were to be performed. Thus I consider the current work redundant, it may be a small part of a different paper in materials and methods, but overall, obtaining the reference genes from transcriptomic data is a better choice.

Round 2

Reviewer 4 Report

I consider the work redundant but primer design may be a value for the manuscript. Bioinformatic pipes can be performed in complete packages in R making them straight forward